

# Clinician motivational interviewing skills in 'simulated' and 'real-life' consultations differ and show predictive validity for 'real life' client change talk under differing integrity thresholds

Alison Bard[1], Lars Forsberg[2], Hans Wickström[3], Ulf Emanuelson[4], Kristen Reyher[1],* and Catarina Svensson[4],*

[1] Bristol Veterinary School, The University of Bristol, Bristol, United Kingdom
[2] MICLab AB, Stockholm, Sweden
[3] MeetMe Psykologkonsult AB, Gothenburg, Sweden
[4] Department of Clinical Sciences, Swedish University of Agricultural Sciences, Uppsala, Sweden
* Joint last author.

Corresponding author
Alison Bard,
alison.bard@bristol.ac.uk

## ABSTRACT

**Background:** Accurate and reliable assessment of clinician integrity in the delivery of empirically supported treatments is critical to effective research and training interventions. Assessment of clinician integrity can be performed through recording simulated (SI) or real-life (RL) consultations, yet research examining the equivalence of these data is in its infancy. To explore the strength of integrity assessment between SI and RL samples in Motivational Interviewing (MI) consultations, this article examines whether Motivational Interviewing Treatment Integrity (MITI) assessments differ between SI and RL consultations and reviews the predictive validity of SI and RL MI skills categorisations for RL client response language.

**Methods:** This study first compared MITI coding obtained in SI and RL consultations for 36 veterinary clinicians. Multilevel models of 10 MITI behaviour counts and four MITI global scores were run using MLwiN 3.02 to assess if a significant difference existed between SI and RL MITI data, with consultation within clinician within cohort (A and B) as nested random effects. Second, we investigated the effect of SI and RL MI skills groupings on rate of RL client response talk using three multivariable regression models. Two Poisson regression models, with random intercepts for farm and veterinarian and offset for number of minutes of the recordings, were estimated in the statistical software R using the package glmmTMB for the two response variables Change Talk and Sustain Talk. A logistic regression model, with the same random intercepts, with the response variable Proportion Change Talk was also estimated using the same package.

**Results:** Veterinary clinicians were less MI consistent in RL consultations, evidenced through significantly lower global MITI Cultivating Change Talk ($p < 0.001$), Partnership ($p < 0.001$) and Empathy ($p = 0.003$) measures. Despite lower objective MI skills groupings in RL consultations, ranking order of veterinary clinicians by MI skills was similar between contexts. The predictive validity of SI and RL MI skills groupings for RL client Change Talk was therefore similar, with significantly more RL client Change Talk associated with veterinarians categorised in the highest

grouping ('moderate') in both SI ($p = 0.01$) and RL ($p = 0.02$) compared to untrained veterinarians in each respective context.

**Conclusions:** Findings suggest SI and RL data may not be interchangeable. Whilst both data offer useful insights for specific research and training purposes, differing contextual MI skills thresholds may offer a more equitable assessment of clinician RL client-facing MI integrity. Further research is needed to explore the applicability of these findings across health contexts.

## INTRODUCTION

Accurate and reliable assessments of clinician integrity in the delivery of empirically supported treatments are at the heart of effective research and training, ensuring clinician adherence and competence can be meaningfully ascertained (*Decker et al., 2013*). Ongoing investigation of assessment criteria for empirically supported treatments is therefore fundamental to understanding how—and to what extent—assessment instruments reflect clinician adherence and competence for the purposes of training, research and practice. Investigation of assessment criteria additionally enables researchers to explore and evidence the theorised links between in-session clinician behaviour and client outcomes.

Motivational Interviewing (MI)—a change-oriented, evidenced-based communication methodology—is one such empirically supported treatment, with extensive research adopting the Motivational Interviewing Treatment Integrity (MITI) code (*Moyers, Manuel & Ernst, 2014*) as a measure of clinician MI skill. The MITI is utilised within research and training contexts where rigor in supervision and evaluation is required (*Moyers et al., 2016*), as a marker for training efficacy (*Chéret et al., 2018*), skills proficiency (*Dunn et al., 2015*), sequential clinician-client linguistic relationships (*Klonek et al., 2016*) and client linguistic and behavioural outcomes (*Svensson et al., 2020a*). The MITI code therefore contributes significantly to how MI is perceived, learned and enacted across disciplines. The gold standard for coding processes is the independent audio coding of recorded clinician sessions where independent raters examine (i) interactions with real life (RL) clients selected from the clinicians' caseloads (*e.g., Forsberg et al., 2010*), (ii) simulated interactions (SI) in which clinician and actor role-play the relevant clinical encounter (*e.g. Sholomskas et al., 2005*) or (iii) both (*Miller et al., 2004*).

Selection of SI or RL client assessment(s) may be based on practical, ethical or theoretical factors. For example, SI offers researchers and trainers control over both role-play actor and consultation scenarios. This allows assessment to be tailored to assess maximum therapist skill, enables standardising and comparability between therapist skill sets and removes the need for informed client consent (*Decker et al., 2013*). RL allows for

assessment of more adaptive clinician proficiency given fluctuating case difficulty, differing client presentations and variable practice settings (*Decker et al., 2013*). In consequence, RL demands more intensive ethical scrutiny with client recruitment and consent. At the heart of this choice also lies a critical empirical consideration: whether clinician skill in SI demonstrates predictive validity for estimating clinician skill with RL clients.

At present, research studies comparing SI and RL consultations *via* the MITI code are in their infancy. *Decker et al. (2013)* found poor agreement on performance criterion and weak associations between SI and RL in clinician adherence and competence ($r = 0.05–0.27$). *Imel et al. (2014)* identified an average relationship of therapist adherence ($r = 0.4$; range 0.04–0.75) between SI and RL and substantially less between-patient variance in adherence scores for SI than RL sessions. Discourse analysis studies also suggest meaningful differences between SI and RL. In SI, relationships may be functionally different, where role-players' knowledge of diagnosis and treatment reverses traditional 'power' roles (*Atkins, 2018*), which may be of relevance to MITI scoring foci such as partnership and collaboration. Clinicians may also make interactional moves that explicitly speak to assessment requirements, with behaviours *'unpacked more elaborately, exaggeratedly or explicitly'* (*Stokoe, 2013*). For instance, clinicians may sometimes exaggerate empathy (*Atkins, 2018*), a core MITI attribute. Accordingly, *Decker et al. (2013)* declared that '*more research is needed to develop the procedures and psychometric strength of integrity assessment based on role-played sessions, including empirically linking benchmarks of performance to client outcomes*'.

Our study attempts to meet this demand, analysing MI skills of veterinary medicine clinicians (hereafter veterinarians) working in cattle herd health advisory services in Sweden using the MITI 4.1 (MITI: *Moyers, Manuel & Ernst, 2014*). Previous analysis of this sample indicated that communication in SI and RL was stylistically similar (*Svensson et al., 2019a*) yet the researchers stated that '*further research is needed to explore if more nuanced differences may exist between such sample groups*'. To explore the strength of integrity assessment between SI and RL samples—by assessing if meaningful differences exist in verbal behaviours between these contexts—this article first examines how MITI integrity assessments differ between SI and RL samples, with a focus on MI skills parameters adopted in training and research categorisation (MITI global scores and behaviour counts). Second, this article reviews the predictive validity of SI and RL MI skills categorisations for RL client response language.

## MATERIALS AND METHODS

### Study participants and data collection timeline

Veterinarians were recruited for a study aiming to evaluate the current communication styles of Swedish large animal veterinarians involved in herd health advisory services and their capacity to learn and integrate MI into their work (hereafter primary study). The selection of participating veterinarians has been described previously by *Svensson et al.*
*(2019a)*. In short, volunteers were selected from the two largest employers of Swedish dairy cattle veterinarians—the Swedish Board of Agriculture and the regional dairy associations—or from among self-employed dairy cattle practitioners involved in the main Swedish network for herd health advisory services. Selection was performed in early 2016, when the total number of dairy cattle veterinarians involved in herd health advisory services in the above three employment groups was estimated to be 97. In total, 42 veterinarians were enrolled in the project; the present study used those 36 veterinarians who remained in the project throughout its full course of 2 years—three men and 33 women (See Table 1 for demographic details). Of those six veterinarians who did not complete the study, reasons included the timeline conflicting with maternity leave ($n = 1$), change in professional employment ($n = 2$) and practical/personal complications that inhibited them engaging with the full study design ($n = 3$). As described, the reasons were not related to MI performance but rather to aspects of participants' personal life, employment and/or workload. The selection of the 170 participating farms has been described previously by *Svensson et al. (2019b)*. In short, farms were a convenience sample chosen by the veterinarians from among their clients. Clients did not receive any compensation to participate in the study. Farmers and their staff were informed about the purpose and design of the project by the veterinarians.

As part of the primary study, the sample of veterinarians was split into two cohorts; one cohort (A: $n = 18$) recorded SI and RL consultations before they received MI training, whereas one cohort (B: $n = 18$) received MI training followed by recording of SI and RL consultations (Fig. 1). Participants were randomized into the two groups at the start of the project, that is before group A veterinarians started their SI recordings in 2016 (see Fig. 1), using a computer-generated random number.

The MI training content and timeline for both groups has been described in detail by *Svensson et al. (2020b)*. In short, MI training consisted of six workshops with theoretical lectures and practical training stretched over a 6 to 7 month time frame. During the time between workshops (3 to 5 weeks), participants were to read and reflect on chapters in the main MI handbook by *Miller & Rollnick (2012)* and to practice their newly learnt skills. They also recorded consultations with clients for use in reflective exercises at the workshops and substantial parts of the workshops were devoted to coaching and feedback on these recordings. Both cohorts of veterinarians received MI training without any cost. We asked veterinarians in both cohorts (A and B) to provide the same information about the communication training so that farms would be blinded to whether the veterinarian had received MI training or not.

## Consultation data

For SI data, each veterinarian took part in three audio-recorded role-play consultations reflecting '*telephone consultations with a client whom the veterinarian previously had met on the farm when the time had been restricted and an agreement therefore had been made to*

**Table 1 Demographic details of veterinary cohort ($n = 36$).**

| Demographic | Veterinary cohort ($n = 36$) |
|---|---|
| **Gender** | |
| Male | 3 |
| Female | 33 |
| **Age** | |
| 30–34 | 9 |
| 35–39 | 10 |
| 40–44 | 4 |
| 45–49 | 2 |
| 50–54 | 6 |
| 55–60 | 5 |
| **Years as a vet** | |
| ≤1 years | 0 |
| ≥1–≤5 years | 8 |
| 5–15 | 16 |
| >15 | 12 |
| **Years as vet in dairy herds** | |
| ≤1 years | 0 |
| ≥1–≤5 years | 10 |
| 5–15 | 17 |
| >15 | 9 |
| **Years in VHHM** | |
| ≤1 years | 4 |
| ≥1–≤5 years | 18 |
| 5–15 | 9 |
| >15 | 5 |

*continue and finish the consultation over telephone'*. SI topics consisted of three common consultation scenarios: (a) increased occurrence of a digestive disorder (displaced abomasum) discussed with a farm owner; (b) udder health problems discussed with a farm manager; and (c) calf diarrhoea problems discussed with a calf caretaker. SI consultations took place from March to May 2016 (cohort A: pre-training session) as well as from March to May 2017 (cohort B: post-training session), with each veterinarian performing three role-plays with an actor experienced in role-play in various settings where MI is used. SI for cohorts A and B involved the same three disease and farm situations (scenarios a–c) but, for practical reasons, not the same actors. Five actors were involved; Actors 1 and 2 (female) participated in the SI for cohort A only; Actor 3 (female) participated in both sessions; Actors 4 and 5 (male) participated in the SI for cohort B only.

As described by *Svensson et al. (2019a)*, actors were not provided with a script, but instead received a farm profile and background information to shape their character. Actors were instructed to be initially ambivalent about the behaviour change in question
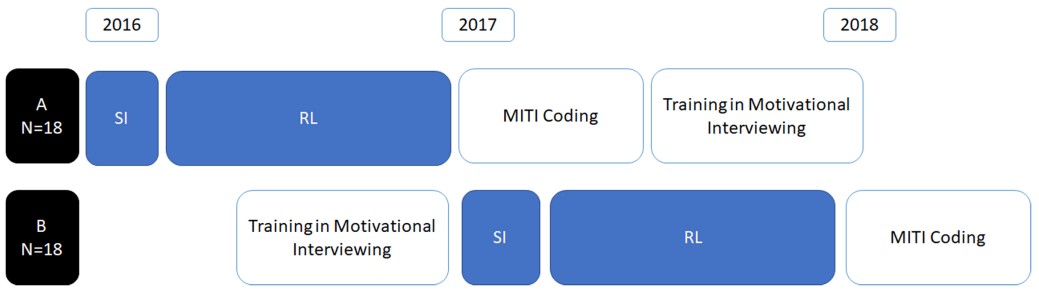

**Figure 1** Veterinarians' (*n* = 36) recordings and subsequent coding by the study timeline, indicating where cohorts recorded SI and RL consultations prior to (A) or after (B) motivational interviewing training. MITI—Motivational Interviewing Treatment Integrity coding system 4.2.1 (*Moyers, Manuel & Ernst, 2014*).

and to respond to the veterinarians' consultation communication in an appropriate and genuine manner given their assumed character, as a means to generate an authentic simulation of the veterinarian-client encounter. The instruction was informed by the MI method that is used in consultations with clients who are ambivalent to support change in a target behaviour. Veterinarians received detailed information *via* e-mail about the farm (including the health problem the consultation was to address and management routines important for this problem) 10 min before they were called by the actor. The target behaviour in these consultations was any preventative action that may improve herd health through addressing the specific health challenge in question.

For RL data, participating veterinarians were asked to record five consultations of veterinary advisory dialogue. Inclusion criteria, related to aims of the primary study, included the need to be able to specify the preventive actions discussed during the visit in a document (health plan) and that the farm had an interest in following up this health plan during a later visit. Veterinarians discussed their participation in the project with farm clients and engaged their participation. All farm clients provided written consent for their involvement and for the use of audio recordings of their consultations for research purposes (Regional Ethical Review Board in Uppsala: reference number 2016/41). In cases of longer consultations, veterinarians could choose whether to record (i) the 20 min considered most relevant in their advisory dialogue or (ii) the whole consultation, informing the coding laboratory of which parts they considered most relevant.

### Assessment of MI skills and coding of client language

Veterinarians' MI skills were assessed by professional coders at MIC Lab AB (Stockholm) in accordance with the MITI 4.2.1 manual (*Moyers, Manuel & Ernst, 2014*), translated into Swedish. Coders did not know the identities of veterinarians, veterinary clients or actors. The MITI 4 coding manual (*Moyers, Manuel & Ernst, 2014*) identifies frequency counts of 10 verbal behaviours, assessments of four global scores on a Likert scale ranging from 1 ("low") to 5 ("high") based on 20 min of the conversation and six summary measurements (Supplemental Materials 1).

The coding of client verbal responses adopting the Client Language Easy Rating (CLEAR) coding system (*Glynn & Moyers, 2012*: Supplemental Materials 2) has previously been described in *Svensson et al. (2020a)*. In short, three coders performed all CLEAR coding of the 170 audio recordings from RL on-farm consultations, according to the CLEAR manual translated to Swedish (*Glynn & Moyers, 2012*). SI client responses were not analysed. We summarized RL client talk as Change Talk and Sustain Talk. We also calculated another outcome variable, Proportion Change Talk, defined as Change Talk frequency over the sum of Change Talk frequency plus Sustain Talk frequency (%CT = CT/(CT + ST)). Coders started CLEAR-coding the RL recordings when all veterinarians (from both cohorts) had recorded all their consultancies. The order in which coders coded the recordings was randomized so that consultancies from both cohort A and cohort B were coded in parallel.

To sustain coders' competence, coders at MIC Lab meet every week for 2-h training sessions and the inter-rater reliability between coders for MITI coding are calculated and checked. Tests adjacent to the codings in the present study were performed in June 2017 and June 2018 and found the intra-class correlations of the different MITI variables to be 0.61–0.97 and 0.52–0.93, respectively.

## Data editing

To ensure equitable data comparison between consultation contexts, the frequency of MITI and CLEAR behaviour counts for conversations shorter than 20 min were adjusted to 20 min. This was achieved by multiplying the counts with 20/(number of minutes of the recordings).

For the purpose of assessing the predictive validity of MI skills thresholds, veterinarians' SI samples were classified by MI skills on a graded scale of 'Poor', 'Near Moderate' or 'Moderate' based on three of the 20 MITI variables (Relational, Cultivating Change Talk and MI Non-adherent behaviours; Table 2). Choice of variables was informed both by the expertise and experience of the research team, in addition to evidence of these variables being theoretically meaningful and correlated with behaviour outcomes (*Lindqvist et al., 2017*; *Romano & Peters, 2016*). Variable adoption of MI skills within the trained veterinary sample (*Svensson et al., 2020b*) meant MITI thresholds of 'fair' and 'good' skill based on expert opinion regarding traditional contexts of MI use (*Moyers et al., 2016*) would not have allowed for meaningful distribution of veterinarians for analysis between thresholds. Veterinarians' RL MI skills were similarly classified as 'Poor', 'Near Moderate' or 'Moderate' based on two of the 20 MITI variables (Relational and Cultivating Change Talk; Table 2). RL MI skills thresholds were set at an objectively lower level than SI thresholds to allow for meaningful distribution of veterinarians between MI skills groupings for analysis, given a disparity in skills in RL compared to SI.

Previous assessment (*Svensson et al., 2020b*) indicated no veterinarians in cohort A reached skills comparable to 'Near Moderate' skills grouping (either in SI nor in RL recordings pre-MI training). On the possibility that 'Poor' skilled veterinarians in cohort A and cohort B might differ in ways that impacted a client (not captured with the MITI integrity assessment), the addition of training was included to further delineate

**Table 2 Motivational Interviewing Treatment Integrity (MITI) skills thresholds established for veterinarians in simulated interactions and real-life consultations.**

| Variable | Simulated interaction | | Real life | |
| --- | --- | --- | --- | --- |
| | Near moderate | Moderate | Near moderate | Moderate |
| Relational global score | >3.5[a] | >3.5[a] | 2 < 3[d] | >3[d] |
| Cultivating Change Talk global score | >2.7[c] | >3[a] | 1.2 < 2[d] | >2[d] |
| MI Non-adherent verbal behaviour count | <4[c] | <2[b] | n/a[d] | n/a[d] |

Notes:
[a] Reflects threshold suggested by *Moyers, Manuel & Ernst (2014)*.
[b] Reflects threshold based on *Miller & Rollnick (2012)*.
[c] Reflects threshold chosen to form a meaningful difference from that of 'Moderate' based on experience from MITI coding in different contexts.
[d] Reflects thresholds allowing RL graded skills groupings (see: Methods).
MITI, Motivational Interviewing Treatment Integrity coding system 4.2.1 (*Moyers, Manuel & Ernst, 2014*); Relational global, (Partnership + Empathy)/2; MI Non-adherent verbal behaviour, Persuade + Confront; n/a, not applicable.

veterinarian MI skills groupings. This created four groups for analysis: 'No training: Poor', 'Training: Poor', 'Training: Near Moderate' and 'Training: Moderate'.

## Statistical analysis

Descriptive statistics (mean, standard deviation) for SI and RL data were completed using IBM SPSS Statistics 24. Multilevel models of the 10 MITI behaviour counts and four MITI global scores were run using MLwiN 3.02 to assess if a significant difference existed between SI and RL MITI data, with consultation within veterinarian within cohort (A and B) as nested random effects (to control for any influence of individual veterinarian and/or their training cohort allocation). Negative binomial models were used to analyse behaviour counts of Giving Information, Question, Persuade, Persuade with Permission, Simple Reflection, Complex Reflection, Affirmation and Seeking Collaboration. Linear regression models were used to analyse globals of Cultivating Change Talk, Partnership and Empathy, while logistic regression models were used to analyse the global Softening Sustain Talk and behaviour counts Confront and Emphasise Autonomy. Adherence to assumptions for the linear and negative-binomial regression models was checked by graphically assessing the residuals; all models showed satisfactory fit.

Details of MITI-CLEAR analysis have been reported previously (*Svensson et al., 2020a*). In short, we investigated the effect of MI skills on rate of client response talk using three multivariable regression models. Two Poisson regression models, with random intercepts for farm and veterinarian and offset for number of minutes of the recordings, were estimated in the statistical software R (*R Core Team, 2019*) using the package glmmTMB for the two response variables Change Talk and Sustain Talk. A logistic regression model, with the same random intercepts, with the response variable Proportion Change Talk was also estimated using the same package. The models also adjusted for a number of extra explanatory variables regarding the veterinarian (gender, type, age, education and experience of advisory work), the client (role on the farm, number of participants from the farm, satisfaction with the consultation), and the consultation (gender concordance, if

**Table 3 Descriptive statistics of MITI global scores and verbal behaviours of 36 Swedish dairy cattle veterinarians in simulated interactions ($n$ = 106) compared to real life ($n$ = 170) consultations.**

| MITI component | Variable | Simulated interaction | | Real life | |
|---|---|---|---|---|---|
| | | *M* | *SD* | *M* | *SD* |
| Global score | Cultivating Change Talk | 2.63 | 0.79 | 1.58 | 0.75 |
| | Softening Sustain Talk | 3.58 | 0.59 | 3.27 | 0.72 |
| | Partnership | 3.12 | 0.93 | 2.05 | 0.89 |
| | Empathy | 2.95 | 0.96 | 2.27 | 1.00 |
| Verbal behaviour | Giving Information | 12.45 | 5.75 | 17.1 | 8.15 |
| | Persuade | 6.37 | 5.38 | 5.29 | 4.05 |
| | Persuade with Permission | 3.37 | 3.02 | 0.99 | 1.40 |
| | Question | 9.84 | 5.66 | 15.46 | 10.66 |
| | Simple Reflection | 2.53 | 1.88 | 5.61 | 5.27 |
| | Complex Reflection | 3.13 | 2.54 | 4.61 | 4.55 |
| | Affirmation | 1.77 | 1.78 | 2.52 | 2.47 |
| | Seeking Collaboration | 2.82 | 1.98 | 0.62 | 0.99 |
| | Emphasise Autonomy | 0.31 | 0.61 | 0.11 | 0.39 |
| | Confront | 0.13 | 0.52 | 0.14 | 0.49 |

Note:
MITI, Motivational Interviewing Treatment Integrity coding system 4.2.1 (*Moyers, Manuel & Ernst, 2014*); *M*, mean; *SD*, standard deviation; Global Score, Likert data 1–5; Verbal Behaviour = continuous data (frequency count).

both client and veterinarian felt the time allocated was sufficient, visit type). Model fit was assessed by graphical examination of randomized quantile residuals and by using tests of under- and over-dispersion and zero inflation, none of which revealed any deviations. No evidence of multicollinearity was found.

## Ethics statement

The study was granted ethics approval by the Regional Ethical Review Board in Uppsala (reference number 2016/041), ensuring procedures met ethical guidelines for research with human participants. Participation in the study was voluntary both for farms and veterinarians. Veterinarians informed their clients about the purpose and methods of the study. Both veterinarians and farm owners and staff provided written consent for sharing data from recordings with the research team. Participants were assured that all information would be treated anonymously and that they could withdraw from the study at any time. They were also assured that data would be stored at the Swedish University of Agricultural Sciences and that no unauthorized person would be able to access the data.

## RESULTS

### MI integrity assessments in SI and RL contexts

Descriptive statistics of MITI variables in SI and RL consultations are shown in Table 3. Veterinarians scored significantly lower in global measures of Cultivating Change Talk ($p < 0.001$), Partnership ($p < 0.001$) and Empathy ($p = 0.003$) in their RL consultations than in their SI consultations. In their RL consultations, they also offered significantly less

**Table 4 Regression analysis of 36 Swedish dairy cattle veterinarians' MITI global scores and verbal behaviours in simulated interactions ($n$ = 106) compared to real life ($n$ = 170) consultations.**

| MITI component | Variable | Fixed effect | 95% CI | | $p$ |
|---|---|---|---|---|---|
| | | | LL | UL | |
| Global score | Cultivating Change Talk[a] | −1.08 | −1.3 | −0.86 | <0.001 |
| | Softening Sustain Talk[b] | −0.75 | −1.65 | 0.16 | 0.10 |
| | Partnership[a] | −1.07 | −1.48 | 0.67 | <0.001 |
| | Empathy[a] | −0.70 | −1.16 | 0.24 | 0.003 |
| Verbal behaviour | Giving Information[c] | 0.33 | 0.06 | 0.59 | 0.02 |
| | Persuade[c] | −0.19 | −0.44 | 0.07 | 0.15 |
| | Persuade with Permission[c] | −1.22 | −1.51 | −0.93 | <0.001 |
| | Question[c] | 0.45 | 0.21 | 0.69 | <0.001 |
| | Simple Reflection[c] | 0.81 | 0.55 | 1.06 | <0.001 |
| | Complex Reflection[c] | 0.39 | −0.30 | 1.08 | 0.27 |
| | Affirmation[c] | 0.35 | 0.09 | 0.62 | 0.009 |
| | Seeking Collaboration[c] | −1.52 | −1.97 | −1.06 | <0.001 |
| | Emphasise Autonomy[b] | −1.14 | −1.82 | −0.46 | 0.001 |
| | Confront[b] | 0.11 | −0.74 | 0.97 | 0.80 |

**Notes:**
[a] Reflects linear regression model.
[b] Reflects logistic regression model.
[c] Reflects negative binomial regression model.
MITI, Motivational Interviewing Treatment Integrity coding system 4.2.1 (*Moyers, Manuel & Ernst, 2014*); CI, confidence interval; LL, lower limit; UL, upper limit; $p$, $p$ value; Global Score, Likert data 1–5; Verbal Behaviour, continuous data (frequency count).

Seeking Collaboration ($p < 0.001$) and Persuade with Permission ($p < 0.001$), whilst asking significantly more Questions ($p < 0.001$), engaging significantly more in Giving Information ($p = 0.02$) and offering significantly more Simple Reflections ($p < 0.001$) and Affirmations ($p = 0.009$) compared to SI. Veterinarians did not differ significantly in their use of Persuade, Confront or Complex Reflection between RL and SI (Table 4).

## Clinician MI skills and client response language

In both SI and RL consultations, veterinarians' grouping by relative MI skills was associated with RL client Change Talk. RL clients whose veterinarians were grouped as 'Training: Moderate' in SI assessment expressed 1.55 times more Change Talk ($p = 0.01$) than RL clients whose veterinarians were grouped as 'Untrained: Poor' in SI assessment. RL clients whose veterinarians were grouped as 'Training: Moderate' in RL assessment expressed 1.48 times more Change Talk ($p = 0.02$) than RL clients whose veterinarians were grouped as 'Untrained: Poor' in RL assessment. No associations between MI skills assessment and Sustain Talk or Proportion Change Talk were detected (Table 5). Ranking order of veterinarians within the four skills groups (*i.e.*, 'No training: Poor', 'Training: Poor', 'Near Moderate', 'Moderate') were comparable between SI and RL contexts, with five of the 36 veterinarians being placed in a different skill grouping in SI and RL.

**Table 5 Effect of veterinarian MI skills grouping on client response language within RL ($n$ = 170) consultations, when skills grouping is derived from either veterinarian SI or RL MITI skills categorisation.**

| Client response language | Variable Overall $p$ for MI skills variable | Factor Level (MI skills) | Effect | CI LL | UL | $p$ | Overall $p$ for MI skills variable | Factor Level (MI skills) | Effect | CI LL | UL | $p$ |
|---|---|---|---|---|---|---|---|---|---|---|---|---|
| | | **Simulated interaction** | | | | | | **Real life** | | | | |
| Change Talk[a] | MI skills (Chi.sq. $p$ = 0.059) | No training: Poor ($n$ = 18) | Ref. | | | | MI skills (Chi.sq. $p$ = 0.081) | No training: Poor ($n$ = 18) | Ref. | | | |
| | | Training: Poor ($n$ = 7) | 0.99 | 0.70 | 1.40 | 0.97 | | Training: Poor ($n$ = 6) | 0.95 | 0.67 | 1.30 | 0.77 |
| | | Training: Near Moderate ($n$ = 5) | 1.07 | 0.77 | 1.50 | 0.68 | | Training: Near Moderate ($n$ = 6) | 1.21 | 0.87 | 1.70 | 0.26 |
| | | Training: Moderate ($n$ = 6) | 1.55 | 1.12 | 2.10 | 0.01 | | Training: Moderate ($n$ = 6) | 1.48 | 1.07 | 2.00 | 0.02 |
| Sustain Talk[a] | MI skills (Chi.sq. $p$ = 0.506) | No training: Poor ($n$ = 18) | Ref. | | | | MI skills (Chi.sq. $p$ = 0.388) | No training: Poor ($n$ = 18) | Ref. | | | |
| | | Training: Poor ($n$ = 7) | 1.08 | 0.71 | 1.60 | 0.73 | | Training: Poor ($n$ = 6) | 1.06 | 0.71 | 1.60 | 0.78 |
| | | Training: Near Moderate ($n$ = 5) | 1.05 | 0.72 | 1.50 | 0.80 | | Training: Near Moderate ($n$ = 6) | 1.05 | 0.72 | 1.50 | 0.81 |
| | | Training: Moderate ($n$ = 6) | 1.35 | 0.92 | 2.00 | 0.13 | | Training: Moderate ($n$ = 6) | 1.38 | 0.95 | 2.00 | 0.09 |
| Proportion Change Talk[b] | MI skills (Chi.sq. $p$ = 0.877) | No training: Poor ($n$ = 18) | Ref. | | | | MI skills (Chi.sq. $p$ = 0.756) | No training: Poor ($n$ = 18) | Ref. | | | |
| | | Training: Poor ($n$ = 7) | 0.93 | 0.63 | 1.40 | 0.72 | | Training: Poor ($n$ = 6) | 0.88 | 0.60 | 1.30 | 0.53 |
| | | Training: Near Moderate ($n$ = 5) | 1.02 | 0.71 | 1.40 | 0.93 | | Training: Near Moderate ($n$ = 6) | 1.14 | 0.79 | 1.60 | 0.48 |
| | | Training: Moderate ($n$ = 6) | 1.12 | 0.79 | 1.60 | 0.51 | | Training: Moderate ($n$ = 6) | 1.07 | 0.76 | 1.50 | 0.69 |

Notes:
[a] Reflects Poisson regression.
[b] Reflects logistic regression.
MI, Motivational Interviewing; Proportion Change Talk, Change Talk frequency/(Change Talk frequency + Sustain Talk frequency); CI, confidence interval; LL, lower limit; UL, upper limit; $p$, $p$-value.

# DISCUSSION

## MI integrity assessments in SI and RL contexts

Results revealed no clear congruence between MI skills in SI and RL contexts for veterinarians engaged in advisory consultations, reflecting previous data from clinicians in human medicine (*Decker et al., 2013*; *Imel et al., 2014*). For the global parameters of Cultivating Change Talk, Partnership and Empathy, veterinarians expressed more affinity

for an MI communication style in SI. Veterinarians focused more on evoking the client's own language in favour of and confidence for change, conveyed more readily that they perceived that the expertise and wisdom for change resided with the client and made more explicit efforts to grasp client perspective and experience (*Moyers, Manuel & Ernst, 2014*: Supplemental Materials 1). Results therefore suggest that there was a meaningful difference in the gestalt approach to client interaction between the SI and RL contexts.

It is possible that the observed differences in performance may represent differences in the complexity of client presentation. Previous literature (*Imel et al., 2014*) has identified substantially less between-patient variance in adherence scores for SI than RL sessions, hypothesized to result from clinicians being more able to deliver consistent, comparable skills in the face of standardised client presentations. Indeed, as a result of those findings, SI may be considered a more equitable assessment of clinician practice (*Liness et al., 2019*). However, in the present study the significant global parameters of Cultivating Change Talk, Partnership and Empathy showed very similar levels of data dispersion in SI and RL interactions (Table 3) suggesting that consistency of skills were relatively equal between consultation contexts.

The observed difference between SI and RL contexts may instead stem from the traditional communication approach associated with veterinary advisory processes/settings, where veterinarians act as the expert imparting instruction (*VetFutures Project Board, 2015*) in a paternalistic manner (*Bard et al., 2017*; *Svensson et al., 2019a*). It is possible that a paternalistic communication approach may more readily be evoked in RL conversations where clinician and client have an established working relationship. Indeed, *Hanna & Fins (2006)* argued that in SI, traditional authoritative relationships are in fact inverted given that '*knowledge and judgement rest with the simulation patient*' (p. 266) through their ability to deliberately steer the course of improvisation and be emotionally detached from their clinical presentation. In human medicine, the paternalistic communication approach is well recognised as endemic to many healthcare systems (*Coulter, 1999*; *Roter, 2000*), meaning similar changes may be witnessed in how SI unfold following suspension of real working relationships and an inversion of authority (*Hanna & Fins, 2006*). This may thus be an important relational consideration in assuming congruence between SI and RL for research and training purposes across medical contexts.

Given the hypothesis of paternalistic communication being more prominent in RL than in SI, it is surprising to observe that no significant difference in Persuade was witnessed between SI and RL. However, Persuade is only one element of a paternalistic communication style, and other results (the significantly lower levels of Cultivating Change Talk, Partnership, Empathy, Persuade with Permission and Seeking Collaboration in RL) align well with this hypothesis. The significantly higher use of questions in RL—and thus a potentially more evocative interaction—could also contest the hypothesised paternalistic paradigm. However, the lower RL skill in Cultivating Change Talk suggests questions were aimed at information seeking rather than eliciting the thoughts and feelings of the client regarding change, as previously reported by *Bard et al. (2017)*.

The differences observed between the RL and SI contexts in the present study may also be due to differences in gender distribution in the two samples. Doctor-patient dyads

where there is gender concordance (*i.e.* male-male or female-female dyad) evidence greater communication equality and client centredness between doctors and patients, compared to gender discordant dyads (*i.e.* male-female or female-male dyad) (*Sandhu et al., 2009*). In the veterinary context, male-male gender concordance was found to enhance relationship centredness in advisory communication by large animal veterinarians (compared to male-female pairings: *Ritter et al., 2018*). In the present study, in RL data there was concordance in 75 consultations (44%) and discordance in 95 consultations (56%), whilst in SI data there was concordance in 87 consultations (81%) and discordance in 20 consultations (19%); *i.e.* in SI, veterinarians were engaged with clients of the same gender for a higher percentage of interactions, likely a result of high female participant numbers ($n = 33$) compared to male ($n = 3$) combined with the female:male ratio of role-play scenarios (4:2) increasing female-female concordance in this context. This factor may have served to increase the potential expression of MI-relevant client-centered verbal behaviour(s) in SI consultations.

The contrast between consultation complexity may also have been meaningful for differences in communication between these contexts. In SI, only one herd health topic area was discussed in the consultation, for which a behaviour change goal and advisory recommendations could be clearly defined. In contrast, RL consultations may have involved multiple topic areas per consultation discussed in tandem, increasing the complexity of identifying an appropriate consultation focus, target behaviour change and related recommendations. Veterinarians in RL may also have been balancing other attentional demands when selecting and targeting advice, such as distracting farm activities, interpreting pathophysiology and/or reviewing herd health data outputs. Compared to SI, RL consultations may therefore have made it more difficult for veterinarians to focus on and present their 'best' communication skill set. These considerations are also likely to be pertinent in human health contexts, where SI (by design) similarly divorces clinicians from the complexities engendered in real client engagement *in situ*.

Differences between SI and RL found may also have be due to the artificial nature of the SI bringing the assessment of communication to the forefront of the interaction. Health professionals in general practice, under assessment through SI, were noted by *Atkins (2018)* to have exhibited communication behaviours that reflected perceived assessment requirements; person-centred care and empathy were specifically mentioned to be included as exaggerated features of SI discourse. This concern was echoed by *Stokoe (2013)* who found police interview role-plays similarly contained behaviours '*unpacked more elaborately, exaggeratedly or explicitly*' in response to assessment (*Stokoe, 2013*). It is possible that RL consultations, being more authentic, encourage clinicians to suspend awareness of being assessed more readily than in SI. This would in turn reduce clinicians' emphasis on expressing their 'best' communication skill set.

The observed differences between these consultation contexts may also have been influenced by the medium of communication, given interaction differences engendered in consulting a client in person (as in RL samples) or on the phone (as in SI samples). For example, *Kraus (2017)* identified that voice-only communication enhanced empathic

ability, where vocal cues alone provided clinicians with more accurate emotional information than a facial or bodily state, which can sometimes be inconsistent with internal states or used to actively mislead. Further research comparing in person and phone-based consultation paradigms is needed to understand the potential impact of communication medium on the broad range of MITI measures.

Overall, study data suggest SI and RL clinician data are not interchangeable. Differences between SI and RL assessment may be influenced by a variety of factors, from relational attributes of the interaction to the recording context and medium. From a research perspective, further work isolating and examining the relative importance of these features (across professional environments) is a critical next step in understanding how SI and RL differ. This would allow optimal selection of SI and RL samples for integrity assessment processes. From an implementation perspective, the choice of SI or RL for assessment of clinician skill may depend on the intended insight from use of the MITI code assessment. For the goal of understanding whether and how much skill has been learnt (for example, assessing a training intervention), SI may be a legitimate environment to estimate clinicians' 'best' communication skills. For the purposes of MI skills validation (for example, ensuring a 'threshold' is maintained for ongoing interactions with clients), RL may offer a more valid assessment of what level of client-facing MI skill is practically achieved by clinicians. The usefulness of RL skills validation would be predicated on clinicians meeting a representative sample of relevant clients and contexts to control for within therapist heterogeneity, ideally with continual assessments over a set time period (*e.g.* every fifth client) rather than single assessments carried out at fixed time intervals (*Dunn et al., 2016*).

## Clinician MI skills and client response language

Despite differing absolute MI skills thresholds, the predictive validity of SI and RL MI skills grouping for RL client Change talk was similar given congruence in veterinary rankings between contexts. RL clients whose veterinarians were grouped as 'Training: Moderate' in SI assessment expressed significantly more Change Talk in RL samples than clients whose veterinarians were grouped as 'Untrained: Poor' in SI assessment. RL clients whose veterinarians were grouped as 'Training: Moderate' in RL assessment expressed significantly more Change Talk in RL samples than RL clients whose veterinarians were grouped as 'Untrained: Poor' in RL assessment. Client CLEAR data aligned with the technical hypothesis of MI, that more MI-consistent within-session clinician behaviour will reinforce, deepen and increase client language in favour of change (*Romano & Peters, 2016*) compared to less MI-consistent within-session clinician behaviour.

In training and research, the question of what is 'good enough' MI practice is a complex one, with research being unable to define clear clinician skills thresholds (*Magill et al., 2018*). If the intention is to verify clinicians' ability to engage RL clients in effective change-oriented consultations, MITI-CLEAR associations in the present study suggest MITI thresholds may convey more clinically relevant information for training and research if adjusted to be objectively higher in SI than RL assessment. If instead the intention is meeting or exceeding an absolute threshold of MI skills to be classed as 'sufficiently skilled' in MI—such as

entering a professional network through recording submission to the *Motivational Interviewing Network of Trainers* (motivationalinterviewing.org)—one absolute threshold of MI skills, applicable to all aspiring clinicians, intuitively establishes a more equitable assessment measure. However, if for any reason applicants are unable to record or submit SI samples, it is possible they may be at a disadvantage in accessing these networks given question posed on the interchangeability of these data in both this study and wider research (*Decker et al., 2013*, *Imel et al., 2014*).

Despite clear differences in absolute MITI skill levels between SI and RL contexts, ranking of veterinarians by skill was comparable between contexts with only five of the 36 veterinarians being placed in a different MI skills grouping in SI compared to RL assessment (*i.e.* 'No training: Poor', 'Training: Poor', 'Training: Near Moderate' and 'Training: Moderate'). SI may therefore offer predictive validity with regards to estimating clinician skill following a training experience, where clinicians performing better in SI may also be likely to perform better in RL consultations. SI ranking could therefore be a useful means of targeting support and coaching to those most in need following an MI training prior to RL client contact.

What is of note in these data is the significantly higher RL client Change Talk in response to veterinarians reaching the RL 'Training: Moderate' threshold compared to the 'No training: Poor' skills grouping. This was surprising, as veterinarians in RL 'Training: Moderate' had objectively lower global scores (Relational and Cultivating Change Talk: Table 2) when compared to suggested MITI skills thresholds based on expert opinion from use of MI in traditional contexts (*Moyers, Manuel & Ernst, 2014*). In consequence, data suggest more nuanced and gradated MITI skills thresholds in RL compared to SI may be useful, if the intention is to identify which clinicians may be making a meaningful impact on client outcomes. Further examination of the relationship between clinician RL MI skills and client linguistic response is needed in a variety of health disciplines to examine if this relationship is unique to this veterinary context.

### Methodological considerations

MITI coding is a complex process and the coders of these data had not previously had any experience of coding veterinary advisory consultations. Coders may therefore not have fully understood the context, may have misinterpreted situations and may have miscoded or missed codes. To guard against this, the same three coders were used for all consultations and inter-rater reliability was examined and found to be fair to excellent (*Cicchetti & Sparrow, 1981*).

For MITI coding, it is also reasonable to hypothesise that identifying and coding according to a change goal may have been more straightforward for the SI consultations, given that change goals were clearly identified in the construction of the role-play scenarios and coders will have coded SI representing the same three scenarios for every veterinarian in cohorts A and B. In contrast, RL consultations would have been more variable and often several different change topics were discussed in tandem. It is not clear whether this would have (i) made the coding of RL consultations more generous, given that this sometimes necessitated a broader change goal in the coding process (for example, the broad goal of

'improving herd health' in a RL consultation, rather than the narrow focus of 'tackling calf scour' in the calf health in SI) or (ii) made the coding of RL consultations less generous, where less familiarity with the change topic(s) might have caused coders to miss or misinterpret change content and/or language.

Although this sample ($n$ = 36) represents over a third of all available dairy cattle veterinarians in Sweden involved in herd health advisory services in 2016, it is still small in relative terms for quantitative examination. Future research utilising larger sample sizes would offer greater confidence in the outcomes of this work, not only to circumvent known issues associated with smaller sample sizes (*e.g.* risk of selection bias, representativeness and generalisation, susceptibility to outliers) but also to allow a more nuanced examination including structural and demographic factors that may influence these discourse interactions. Additionally, a larger sample with greater variation in distribution of MITI scores across veterinarians may identify further associations between veterinarians' MI skills and client response talk that add to and inform the observations of this study; within this sample, the limited spread of MI skills (few veterinarians reached moderate skills and none reached higher levels of MI skills) may have made it less possible to identify associations with client response talk when examining the MITI-CLEAR relationship.

The authors also note that the differences between SI and RL were measured by combining two separate cohorts: one untrained in MI (cohort A, $n$ = 18) and one trained in MI (cohort B, $n$ = 18). Given the possibility of skills diminishing over time in the MI-trained group (*Miller & Rollnick, 2012*), differences in SI and RL may have been shaped by the trained group's SI samples being recorded closer in time to the training experience, as compared to their RL samples. This possibility led to the insertion of 'training cohort' as a random effect in the multilevel models. Descriptive statistics by cohort (Materials S3) suggested only Persuade with Permission may have been impacted in its magnitude (but not direction) of SI-RL difference for the trained cohort post-training, underpinning confidence in the SI and RL data relationships presented in this article.

## CONCLUSION

Data suggest meaningful differences in SI and RL MI integrity data may exist and that data in each consultation context may not be interchangeable. Differing contextual MI skills thresholds may therefore offer a more equitable assessment of clinician client-facing MI integrity. Furthermore, data suggest that clinicians grouped according to higher objective MI skills thresholds in SI compared to RL consultations evidence comparable predictive validity for RL client Change Talk. Differing contextual MI skills thresholds may therefore also be more predictive of client responses to clinicians within RL MI consultations. From an implementation perspective, both SI and RL as a measure of clinician MI integrity offer useful insights but should best be considered on a case-by-case basis depending on research or training goal(s). Further research is needed to explore the validity and applicability of these findings across diverse health contexts.

## ACKNOWLEDGEMENTS

The authors would like to wholeheartedly thank the participating veterinarians and farmers who gave their time and effort to support this research. The authors are grateful to Åsa Karlin, Nanny Nilsson, Emilia Roosmann, Daniel Ohlsson and Martin Preisler at MIC Lab AB (Stockholm) for their role-play acting performances and to Mahlena Wiveson, Helena Chaomar and Marie Illerbrand at MIC Lab AB (Stockholm) for coding the recordings. Johan Glimskog (MIC Lab AB, Stockholm) is acknowledged for data retrieval and Staffan Betnér and Claudia von Brömssen for performing the statistical analyses of MITI-CLEAR data. Data access statement: Participants in this study were assured that all data would be stored at the Swedish University of Agricultural Sciences and that no unauthorized person would be able to access the data. Data are therefore not deposited in an official repository. Please contact the data owner Catarina Svensson for queries regarding data access: catarina.svensson@slu.se.

### Funding

This study was funded by The Swedish Research Council for Environment, Agricultural Sciences and Spatial Planning (942-2015-960). The funders had no role in study design, data collection and analysis, decision to publish, or preparation of the manuscript.

### Grant Disclosures

The following grant information was disclosed by the authors:
The Swedish Research Council for Environment, Agricultural Sciences and Spatial Planning: 942-2015-960.

### Competing Interests

Lars Forsberg is Operations Manager at MICLab AB.

### Author Contributions

- Alison Bard conceived and designed the experiments, analyzed the data, prepared figures and/or tables, authored or reviewed drafts of the article, and approved the final draft.
- Lars Forsberg conceived and designed the experiments, performed the experiments, authored or reviewed drafts of the article, and approved the final draft.
- Hans Wickström conceived and designed the experiments, performed the experiments, authored or reviewed drafts of the article, and approved the final draft.
- Ulf Emanuelson conceived and designed the experiments, performed the experiments, analyzed the data, authored or reviewed drafts of the article, and approved the final draft.
- Kristen Reyher conceived and designed the experiments, authored or reviewed drafts of the article, and approved the final draft.
- Catarina Svensson conceived and designed the experiments, performed the experiments, authored or reviewed drafts of the article, and approved the final draft.

## Human Ethics

The following information was supplied relating to ethical approvals (*i.e.* approving body and any reference numbers):

The Regional Ethical Review Board in Uppsala granted ethical approval to carry out this research (reference number 2016/41).

## Data Availability

For this original study, the participants only gave consent for data to be stored at the host university (The Swedish University of Agricultural Sciences: SLU) and shared with active researchers connected with the project. These data are recorded consultations between veterinary surgeons and their clients and the coding of these interactions, therefore the personal nature of these data necessitated these restrictions that were established for ethical review.

Participants in this study were assured that all data would be stored at the Swedish University of Agricultural Sciences and that no unauthorized person would be able to access the data. The data are therefore not deposited in an official repository; they are archived at the Swedish University of Agricultural Sciences. Please contact the Archivis, Patrik Spånning Westerlund, at arkiv@slu.se with any queries regarding data access.

## Supplemental Information

Supplemental information for this article can be found online at http://dx.doi.org/10.7717/peerj.14634#supplemental-information.

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
