# Peer review of "Clinician motivational interviewing skills in ‘simulated’ and ‘real-life’ consultations differ and show predictive validity for ‘real life’ client change talk under differing integrity thresholds"

_PeerJ, doi:10.7717/peerj.14634_

## Round 0.1 · original submission · Major Revisions

Please address comments from each of the two reviewers, and explain how you have modified your manuscript to address these changes.

Reviewer 1 ·

Basic reporting

The manuscript deals with a relevant topic: how can we predict whether a trained clinician is ready to perform MI at a sufficient level in real life situations? The paper is well structured and well written, and will certainly contribute to the knowledge on this topic.
At some points, the reader needs more information to be able to better interpret the findings and the assertions, especially in the ‘Materials and methods’ section, and in the ‘Discussion’ section.

Experimental design

Introduction
1. The Introduction section shows the relevance of the study for the field of MI, leading to a relevant research question.

Material and Methods
2. Could you provide the reader with more information on the attrition? At what moment(s) in the research process did the 6 (of the 42) veterinarians drop out? How can we know if the attrition was or was not selective (e.g. all poor performing on MI)?
3. How were the A- and B-group composed? Was there some randomisation procedure? If so, please provide some information on this procedure. If not, why not, and what other procedure was used?
4. To enable the reader to better interpret the findings, it will be helpful to have more knowledge of the characteristics of the 36 participating veterinarians. Age, years of experience, gender etc., preferably divided into the four groups (Untrained: poor; Trained: poor; Trained: near moderate; Trained: moderate), in order to know the degree of comparability of the four groups. This will also help the reader to better interpret the information in line 247-251.
5. Under ‘Consultation data’, could you inform the reader about the target behaviour in the SI-condition (line 148-151)? If a calf has a diarrhoea problem, I would think that the farmer would already be motivated for (secondary) prevention. And (line 159/160), related to this topic, “the actors were instructed to be initially ambivalent” about what?
6. And the same question on the RL-condition (lines 167-170). Is the target behaviour (to implement) preventive actions?
7. If SI client responses were not analysed (line 189), then how has it been possible to compose the left half of table 4?
8. Could you also provide us with the inter-rater reliability scores on the CLEAR (line 196-200)?
9. Under ‘Data editing’, could you clarify your choice to classify the veterinarians on only three MITI variables (SI-condition, line 206-210) or two MITI variables in the RL-condition (line 214-216)? The justification by referring to only one research paper (Lindqvist et al., 2017. The other reference is not a study report) about a study in a completely different target group doesn’t seems very convincing. Why didn’t you use the six MITI summary measurements?
10. Table 1. Could you explain why the threshold score on ‘Relational global score’ in the SI-condition is the same for both the Near Moderate and the Moderate group?

Validity of the findings

Results
11. Table 4. Could you add the ‘n’ to each of the four groups in both the left and the right half of this table. it is hard to interpret the findings presented in this table without knowing the size of each group.
12. Did the five switching veterinarians perform better of poorer in RL than in SI (line 286)? This may already be clear after adding the group sizes to table 4.

Discussion
13. The paragraph on the gender distribution (line 333-343) puzzles me. There are three male veterinarians and 33 female veterinarians in the sample (line125/126): 8.3% male; 91.7% female. How can this, in the RL-condition, lead to gender concordance in 44% of the consultations?
14. To interpret the assertion made by the authors in line 394-401, the reader needs more information concerning the five switching veterinarians (see also above, under 11 and 12). In these small subgroups, switching of five out of 18 persons can make a meaningful difference.
15. Under ‘Methodological considerations’, I would have expected a reflection on the small sample size (36 participants). I know that it is hard for this kind of research to reach many participants and keep them engaged the complete study time. So, my comment is not about that. It is about the known consequences of this phenomenon, such as the risk of selection bias, representativeness and generalisation; susceptibility to outliers; susceptibility to incomparable subgroups; and also the problem described in line 454-459.

Additional comments

None.

Reviewer 2 ·

Basic reporting

No comment.

Experimental design

No comment.

Validity of the findings

No comment.

Additional comments

The paper is very well written and completely clear. The methods are well described and the approach taken is logical. The study highlights interesting and important differences in the MI skills exhibited by practitioners in simulated interviews vs. real-life consultations.
PeerJ guidelines ask us not to comment (by implication negatively) on the importance of the work, but I would just add that I think the paper will be of a lot of interest to those of us working in MI skills training, especially for the detailed analysis in the Discussion section of the reasons why these differences may exist.
MI skills acquired during training are often not exhibited by practitioners in practice. This study reminds us that it may be not just a matter of skill retention, but that there are other important differences between simulations with actors and real interviews with clients that derive from the tasks themselves, including differential power relations, different expectations of behaviour, etc.
I have very little to add by way of criticism; maybe just a few typos: I think every instance of "predicated" or "predicative" was probably meant to be "predicted", "predictive". And I think line 483 may be missing the word "thank".

---

## Round 0.2 · accepted · Accept

Thank you for the changes you made. Both reviewers now concur that your manuscript is ready for publication.

Reviewer 1 ·

Basic reporting

No comments.

Experimental design

No comments anymore.

Validity of the findings

No comments anymore.

Additional comments

No comments.

Reviewer 2 ·

Basic reporting

No Comment

Experimental design

No Comment

Validity of the findings

No Comment

Additional comments

As before, I am happy for the article to be published.